# Potent Antibacterial Prenylated Acetophenones from the Australian Endemic Plant *Acronychia crassipetala*

**DOI:** 10.3390/antibiotics9080487

**Published:** 2020-08-06

**Authors:** Trong D. Tran, Malin A. Olsson, David J. McMillan, Jason K. Cullen, Peter G. Parsons, Paul W. Reddell, Steven M. Ogbourne

**Affiliations:** 1GeneCology Research Centre, School of Science and Engineering, University of the Sunshine Coast, Maroochydore DC, Queensland 4558, Australia; ttran1@usc.edu.au; 2School of Health and Sports Sciences, University of the Sunshine Coast, Maroochydore DC, Queensland 4558, Australia; Malin.Olsson@research.usc.edu.au (M.A.O.); dmcmill1@usc.edu.au (D.J.M.); 3QIMR Berghofer Medical Research Institute, Locked Bag 2000, PO Royal Brisbane Hospital, Herston, Queensland 4029, Australia; Jason.Cullen@qimrberghofer.edu.au (J.K.C.); Peter.Parsons@qimrberghofer.edu.au (P.G.P.); 4QBiotics Limited, PO Box 1, Yungaburra, Queensland 4884, Australia; Paul.Reddell@qbiotics.com

**Keywords:** *Acronychia crassipetala*, crassipetalonol A, crassipetalone A, prenylated acetophenone, antibiotics, cytotoxicity

## Abstract

*Acronychia crassipetala* is an endemic plant species in Australia. Its phytochemistry and therapeutic properties are underexplored. The hexane extract of the fruit *A. crassipetala* T. G. Hartley was found to inhibit the growth of the Gram-positive bacteria *Staphylococcus aureus*. Following bio-activity guided fractionation, two prenylated acetophenones, crassipetalonol A (**1**) and crassipetalone A (**2**), were isolated. Their structures were determined mainly by NMR and MS spectroscopic analyses. This is the first record of the isolation and structural characterisation of secondary metabolites from the species *A. crassipetala*. Their antibacterial and cytotoxic assessments indicated that the known compound (**2**) had more potent antibacterial activity than the antibiotic chloramphenicol, while the new compound (**1**) showed moderate cytotoxicity.

## 1. Introduction

As a result of geographic isolation and a vast array of geographical and environmental habitats, Australia is one of the most megadiverse countries in the world with 84% of terrestrial plants being classified as endemic species [1]. Analysis of Australian flora from Australian tropical habitats showed these regions are particularly rich in plant diversity and species endemism [2]. Despite accounting for only 0.3% of the Australian continent, Queensland’s tropical rainforests are internationally recognised as one of the global biodiversity hotspots [3]. This unique ecological resource is a distinct and relatively untapped source of novel and new natural products with therapeutic potential [4]. For example, the Queensland tropical native species *Duboisia myoporoides* R. Brown is a source of tropane alkaloids including scopolamine, which is an important precursor for the synthesis of scopolamine butylbromide, an anticholinergic and antispasmodic drug with the brand name Buscopan [5]. Another example is a novel indolizidine alkaloid, grandisine A, from the Queensland rainforest tree *Elaeocarpus grandis* F. Muell, which exhibited binding affinity for the human *δ*-opioid receptor and has been considered as a potential lead for developing an analgesic agent [6]. More recently, an epoxy-tigliane diterpene, tigilanol tiglate (formerly EBC46), from the Queensland tropical endemic species *Fontainea picrosperma* C. T. White, has been discovered and shown to have significant anticancer activity [7]. Tigilanol tiglate has been approved by the European Medicines Authority [8] as a novel canine therapy for mast cell tumours [9] and is currently in clinical trials to assess its potential as a human anticancer therapeutic [10].

With the success of discovering and developing the novel anticancer natural compound tigilanol tiglate from QBiotics’s plant extract library (EcoLogic^TM^), a new drug discovery campaign has been launched to discover potent antibacterial natural products [11]. A hexane extract from the fruit of *Acronychia crassipetala* T. G. Hartley was shown to be active against the Gram-positive (G+ve) bacteria *Staphylococcus aureus*.

The genus *Acronychia* belonging to the family Rutaceae comprises over 40 species, which have a broad distribution from India, China, Malesia, New Caledonia to Australia [12]. Some *Acronychia* spp. have been used in folk medicines of indigenous Asian and Australian populations for the treatments of diarrhoea, asthma, ulcers, rheumatism, fever and parasitic infections [13,14,15]. About 60 compounds including acetophenones, flavonoids, quinoline and acridone alkaloids have been identified from approximately half of the *Acronychia* spp. plants [12,16]. Several compounds have been found with potential biological activities such as acronine, an anticancer agent from *A. baueri* [17]; acrophyllodine, an antiarrhythmic agent from *A. haplophylla* [18]; and acrovestone, an antityrosinase compound from *A*. *pedunculata* [19]. The species *A. crassipetala* is one of the 19 *Acronychia* spp. recognised in Australia [20]. This shrub is endemic to the wet tropical rainforests of northeastern Queensland and found exclusively at altitudes between 400 and 1250 m [20]. So far, chemical investigation of *A. crassipetala* has been limited to the study of essential oil extracted from *A. crassipetala* leaves [20].

This research reports the isolation, structural elucidation, antibacterial and cytotoxic properties of the two prenylated acetophenones from the fruit of *A. crassipetala*, crassipetalonol A (**1**) and crassipetalone A (**2**), of which crassipetalonol A (**1**) was found as a new compound (Figure 1).

## 2. Results and Discussion

Compound **1** was isolated as a yellowish amorphous powder and had a molecular ion peak at (+) *m*/*z* 321.1694 in HR-ESI-MS corresponding to the molecular formula C_18_H_24_O_5_ with seven degrees of unsaturation. The ^1^H-NMR spectrum of compound **1** displayed eight singlets (*δ*_H_ 13.45, 10.97, 6.06, 4.63, 1.75, 1.70, 1.67 and 1.59), two doublets (*δ*_H_ 4.52 and 3.11), and two triplets (*δ*_H_ 5.40 and 5.07) (Table 1). The ^13^C and edited HSQC experiments confirmed **1** had 18 carbons, including 1 ketone carbonyl (*δ*_C_ 204.4), 3 oxygenated quaternary carbons (*δ*_C_ 162.5, 161.5 and 160.7), 4 olefinic quaternary carbons (*δ*_C_ 137.8, 129.9, 107.0 and 102.4), 3 olefinic tertiary carbons (*δ*_C_ 122.9, 119.2 and 91.3), 2 oxygenated methylenes (*δ*_C_ 68.2 and 64.6), 1 methylene (*δ*_C_ 20.9) and 4 methyl groups (*δ*_C_ 25.5, 25.4, 18.0 and 17.5) (Table 1). A spin system from the methylene at *δ*_H_ 3.11 (2H, d, *J* = 7.2 Hz, H-1″) to the olefinic proton at *δ*_H_ 5.07 (1H, t, *J* = 7.2 Hz, H-2″) showed long range HMBC correlations from H-1″ to C-3″ (*δ*_C_ 129.9), and H-2″ to C-4″ (*δ*_C_ 25.4) and C-5″ (*δ*_C_ 17.5), resulting in the assignment of an *iso*-prenyl unit. A relative orientation of the two methyl groups was defined from NOESY correlations of H-1″/H-5″ and H-2″/H-4″. A similar *iso*-prenyl unit was assigned for the second spin system consisting of the oxygenated methylene at *δ*_H_ 4.52 (2H, d, *J* = 6.4 Hz, H-1‴) and the olefinic proton at *δ*_H_ 5.40 (1H, t, *J* = 6.4 Hz, H-2‴). The remaining six olefinic carbons (*δ*_C_ 162.5, 161.5, 160.7, 107.0, 102.4 and 91.3) and four degrees of unsaturation together with HMBC correlations from H-5′ (*δ*_H_ 6.06) to C-1′ (*δ*_C_ 102.4) and C-3′ (*δ*_C_ 107.0) supported the establishment of a penta-substituted benzene ring system. HMBC correlations of H-1″/C-3′ and H-1‴/C-4′ enabled the first and second *iso*-prenyl units to connect to the benzene ring at C-3′ and C-4′, respectively. A hydroxymethyl ketone group was deduced and connected to C-1′ due to a cross-peak correlation from a hydroxymethyl H-2 to C-1 (*δ*_C_ 204.4) and a four-bond correlation from H-5′ to C-1. HMBC correlations of 2′-OH/C-1′, 2′-OH/C-3′, 6′-OH/C-1′ and 6′-OH/C-5′, and NOESY correlations of 2′-OH/H-1″ and 6′-OH/H-5′ confirmed the positions of the two hydroxy groups (*δ*_H_ 13.45 and 10.97) at C-2′ and C-6′, respectively. Therefore, compound **1** was elucidated as 1-(2′,6′-dihydroxy-3′-(3″-methylbut-2″-en-1″-yl)-4′-((3‴-methylbut-2 -en-1‴-yl)oxy)phenyl)-2-hydroxyethan-1-one with a trivial name crassipetalonol A (Figure 2).

Compound **2** was assigned as 1-(2′,6′-dihydroxy-3′-(3″-methylbut-2″-en-1″-yl)-4′-((3‴-methylbut-2‴-en-1‴-yl)oxy)phenyl)ethan-1-one (trivial name, crassipetalone A) by spectroscopic data comparisons with appropriate literature values [21]. This compound was previously identified from the *Euodia lunu-ankenda* T. G. Hartley root bark [22] and the *Urtica dioica* L. nettle leaf [23]. Crassipetalone A was reported to have a fungicidal activity against *Cladosporium cladosporioides* [22].

The two isolated acetophenones were tested for their antibacterial activity towards several ESCAPE pathogens (Table 2 and Appendix A). While crassipetalonol A (**1**) was found to have low or no activity towards the pathogens at the tested concentration of 156 µM, crassipetalone A (**2**) potently inhibited the G+ve bacteria *S. aureus* and *Entercoccus faecium* with minimum inhibitory concentration (MIC)_75_ values of 2.6–20.6 µM. Importantly, compound **2** displayed 2–4 fold more inhibition against *S. aureus* compared to the antibiotic chloramphenicol. Replacing the acetyl in **2** by the hydroxymethyl ketone in **1** reduced potency against the *S. aureus* strains 30-fold. Although activity against fungi and G+ve bacteria has previously been reported for acetophenone and its derivatives [24,25,26], the presence of the phenolic hydroxy groups with acidity resulted in increased biological activity by uncoupling oxidative phosphorylation [27]. Moreover, the hydrophilic/lipophilic balance of the molecule was found to play an important role in the penetration of the antibacterial agent through a bacterial cell surface [24]. A certain degree of lipophilicity produced by the *iso*-prenyl and other substituents in the acetophenone molecule enhanced the antimicrobial activity [26,27]. The higher lipophilicity of compound **2** compared to **1** was predicted by their octanol–water partition coefficient (ClogP) values (4.63 of **2** versus 3.77 of **1**) [28]. Therefore, compound **2** could penetrate more easily through the cell wall and exert its bactericidal activity. This study also revealed that the isolated acetophenones selectively inhibited the growth of the tested G+ve bacteria rather than the Gram-negative (G-ve) ones (Appendix A). These results were in accordance with previous reports of the antibacterial activity of related prenylated acetophenones [26,29,30]. The selective activity of **2** might be related to cell wall disruption or to another specific target present only in G+ve bacteria.

Cytotoxicity of compounds **1** and **2** was evaluated using a panel of five human cell lines including immortalised keratinocyte cells (HaCaT), adult dermal fibroblast cells (HDF), neonatal foreskin fibroblast cells (NFF), immortalised embryonic kidney cells (HEK293) and hepatoma cells (HepG2) (Table 3). The data suggested that crassipetalonol A (**1**) was 1–5 fold less cytotoxic than crassipetalone A (**2**). However, comparing the antibacterial and cytotoxic activities suggested that compound **2** had more potential as an antibiotic than compound **1**. This difference further supported that the acetyl group contributes significantly to the antibacterial property of the *Acronychia*-type acetophenone skeleton. Although the selectivity indices between human cancer cells and bacterial cells of compound **2** ranged from 1 to 5, which is relatively low, its potent inhibition against the growth of *S. aureus* compared to other prenylated acetophenones reported previously [26,29,30,31] warrants further investigation, including in vivo trials to confirm the value of this compound. In addition, compound **2** could be modified using medicinal chemistry approaches with an aim to further improve the activity/toxicity window.

## 3. Materials and Methods

### 3.1. General Experimental Procedures

IR spectra were obtained on a PerkinElmer Spectrum 400 FT-IR spectrometer (Waltham, MA, USA). NMR spectra were acquired on a Bruker Ascend 400 spectrometer (Billerica, MA, USA) equipped with a 5 mm room temperature probe operating at 400 MHz for ^1^H and 100 MHz for ^13^C. ^1^H and ^13^C spectra were referenced to the residual deuterated solvent peaks of DMSO-*d*_6_ at *δ*_H_ 2.50 and *δ*_C_ 39.5 ppm. HR-ESI-MS data were acquired on a Sciex X500R Q-TOF mass spectrometer (Framingham, MA, USA). HPLC purifications were performed on a preparative Agilent 1200 system equipped with a diode array detector and processed by ChemStation software (C.01.07). All solvents used for extraction and chromatography were HPLC grade and the H_2_O used was Mili-Q water.

### 3.2. Plant Material

Fruits of *Acronychia crassipetala* T. G. Hartley (Rutaceae) were sampled from four mature trees growing in lower montane tropical rainforest at Upper Barron, Queensland, Australia, and combined into a single collection for subsequent analysis. Voucher specimens were also collected from each individual tree and held in the QBiotics Limited herbarium (specimen numbers YA1028a to d).

### 3.3. Extraction and Isolation

Fresh *Acronychia crassipetala* fruits (220 g) were ground and sequentially extracted with *n*-hexane (300 × 2 mL), dichloromethane (DCM) (300 × 2 mL), methanol (MeOH) (300 × 2 mL) and water (H_2_O) (300 × 2 mL). The solvents were then evaporated to yield three extracts (hexane, DCM and MeOH). To 10 mg of each extract, 1 mL of DMSO was added to prepare a stock concentration of 10 mg/mL for MIC and MBC assays. The hexane extract showed antibacterial activity in MIC and MBC assays. The hexane extract (320 mg) was loaded onto a C_18_ Kinetex HPLC column (5 µm, 250 × 21.2 mm) and eluted by a linear gradient at a flow rate of 10 mL/min from 35% MeOH/65% H_2_O to 50% MeOH/50% H_2_O for 5 min; 50% MeOH/50% H_2_O to 100% MeOH over 40 min and isocratic with 100% MeOH for 15 min; 8 fractions (7.5 min each) were collected. Fraction 6 displayed the most potent antibacterial activity with MIC_75_ of 12.5 µg/mL and was therefore selected for further purification. Fraction 6 was fractionated on the same Kinetex HPLC column (5 µm, 250 × 21.2 mm) at a flow rate of 10 mL/min using an isocratic program with 35% MeOH (0.1% formic acid (FA))/65% H_2_O (0.1% FA) for 10 min, a linear gradient from 35% MeOH (0.1% FA)/65% H_2_O (0.1% FA) to 100% MeOH (0.1% FA) over 35 min, and isocratic with 100% MeOH (0.1% FA) for 15 min to yield compounds **1** (32 mg, *t*_R_ = 34.0 min) and **2** (58 mg, *t*_R_ = 36.0 min).

Crassipetalonol A (**1**): yellowish amorphous powder; UV (MeOH) *λ*_max_ (log *ε*) 290 (4.26) and 240 (3.83); IR *ν*_max_ 3214, 2916, 1633, 1595, 1427, 1249 and 1088 cm^−1^; ^1^H and ^13^C-NMR data, Table 1 and Appendix A; (+) HR-ESI-MS *m*/*z* 321.1694 [M + H]^+^ (calcd for C_18_H_25_O_5_^+^, 321.1697, Δ−0.9 ppm), Appendix A.

Crassipetalone A (**2***)*: yellowish amorphous powder; UV (MeOH) *λ*_max_ (log *ε*) 289 (4.41) and 240 (4.06); IR *ν*_max_ 3141, 2929, 1638, 1589, 1436, 1293 and 1087 cm^−1^; ^1^H and ^13^C-NMR data, Appendix A; (+) HR-ESI-MS *m*/*z* 305.1745 [M + H]^+^ (calcd for C_18_H_25_O_4_^+^, 305.1747, Δ−0.7 ppm), Appendix A.

### 3.4. Antibacterial Assays

#### 3.4.1. ESKAPE Pathogens

The bacterial strains used in this study were ***E****nterococcus faecium* ATCC 35667, *E. faecium* C15, ***S****taphylococcus aureus* ATCC 25923, *S. aureus* ATCC 29247, ***K****lebsiella pneumoniae* ATCC 13883, *K. pneumoniae* ATCC 12657, ***A****cinetobacter baumannii* ATCC 19606, *A. baumannii* ATCC 17978, ***P****seudomonas aeruginosa* ATCC 10145, *P. aeruginosa* ATCC 49189, ***E****nterobacter aerogenes* ATCC 13048 and *E. cloacae* ATCC 13047; collectively, commonly referred to as the ESCAPE pathogens.

#### 3.4.2. Minimum Inhibitory Concentration (MIC) and Minimum Bactericidal Concentrations (MBC)

MICs and MBCs of plant extracts and fractions were evaluated using standard methodologies carried out in microdilution formats [32,33]. Bacterial isolates were grown aerobically in Mueller Hinton broth (Oxoid) overnight at 37 °C without shaking. After 16 h, the bacteria were centrifuged, the supernatant removed, and the pellet washed in 5 mL of phosphate buffered saline (PBS). The bacteria were then resuspended in Mueller Hinton broth to a concentration of approximately 1 × 10^6^ CFU/mL. One hundred microliters of serially diluted plant extract or fractions (max concentration 500 µg/mL) were added to the wells of a 96-well plate (Nunc™ MicroWell™, Sigma Aldrich, Sydney, New South Wales, Australia) containing 100 µL of bacterial suspension. Negative controls (containing 1% DMSO only) were also loaded. In addition, serially diluted plant extracts/fractions (no inoculum) were set up to determine background. The plates were then incubated for 24 h at 37 °C. Following incubation, relative bacterial growth in experimental wells compared to control wells was determined by the optical density of the solution at a wavelength of 600 nm measured by a PerkinElmer EnSpire Multimode plate reader (Waltham, MA, USA). We used MIC_75_ (75% inhibition), in conjunction with analysis of dose dependency as the major criteria for identifying fractions/extracts of interest. All extracts were tested in triplicate. For MBC assays, extract/bacterial combinations that displayed significant MIC_75_ activity were also plated onto growth media to assess viability. Chloramphenicol was used as a positive control for *E. faecium* 35667, *E. faecium* C15, *S. aureus* 25923, *S. aureus* 29247, *K. pneumoniae* 13883, *K. pneumoniae* 12657, *A. baumannii* 19606, *A. baumannii* 17978, *E. aerogenes* 13048 and *E. cloacae* 13047. Kanamycin was used as a positive control for *P. aeruginosa* 10145 and *P. aeruginosa* 49189. Each compound was assayed in triplicate with at least three biological replicates.

### 3.5. Cytotoxic Assays

#### 3.5.1. Cell Culture and Reagents

HaCaT (immortalised human keratinocytes), neonatal foreskin fibroblasts (NFF) and HEK293 (immortalised human embryonic kidney cells) were cultured in RPMI media supplemented with 10% foetal calf serum (FCS). HepG2 (human hepatocellular carcinoma) were cultured in DMEM media supplemented with 10% FCS. Adult human dermal fibroblasts (HDF—ThermoFisher Scientific, Waltham, MA, USA) were cultured in Medium 106 (ThermoFisher Scientific, Waltham, MA, USA) supplemented with low serum growth supplement (LSGS) and gentamycin. Cells were maintained in a humidified incubator at 37 °C with 5% CO_2_ and passaged using trypsin/versene. All cell lines were confirmed mycoplasma negative prior to use, using MycoAlert (Promega, Madison, WI, USA). HDF and NFF were used between p3 and p10 for all assays.

#### 3.5.2. Cell Growth/Survival Assays

All cells were seeded into clear 96-well plates (Corning #3595, Sigma Aldrich, Sydney, New South Wales, Australia) in 100 μL of media at the following cell concentrations: HaCaT and HEK293, 1000 cells per well; NFF and HDF, 2000 cells per well; and HepG2, 3000 cells per well. After 24 h, the media was removed and 90 μL of fresh media was inserted into each well. Compounds were prepared via serial dilution in the media (to 10× final assay concentration). Vehicle (DMSO)-only controls were also prepared. An amount of 10 μL of compound/vehicle dilutions were subsequently added to cells in duplicate and the resultant plate/s were incubated in a humidified incubator at 37 °C and 5% CO_2_ for either 4 (HaCaT), 5 (HEK293) or 7 (HDF, NFF and HepG2) days. Cell growth/viability was measured in each well using a CellTiter 96^®^ AQ_ueous_ One Solution Cell Proliferation Assay kit according to the manufacturer’s instructions (Promega). Absorbance values were recorded at 490 nm with a H4 Hybrid Synergy plate reader (Biotek). A media-only control was also compiled for background subtraction. Modified absorbance values from compound treated wells were normalised to vehicle-treated samples and the %growth/survival in each sample was determined. The %growth/survival was plotted against Log_10_[Compound] μM to generate absolute IC_50_ curves for each compound using PRISM 6.0. Doxorubicin (Sigma Aldrich, Sydney, New South Wales, Australia) was used as a positive control. Each compound was tested in duplicate with at least three biological replicates.

## 4. Conclusions

In conclusion, two prenylated acetophenones, crassipetalonol A (**1**) and crassipetalone A (**2**), were successfully isolated from the fruit of the Australian endemic plant *A. crassipetala*. The assessments of their biological activities indicated that the new acetophenone (**1**) showed relatively high levels of cytotoxicity, while the known compound (**2**) exhibited relatively high levels of antibacterial activity. With these findings, this study broadens our understanding of the secondary metabolites of the underexplored species *A. crassipetala* and the therapeutic potential of prenylated acetophenones.

## Figures and Tables

**Figure 1 antibiotics-09-00487-f001:**
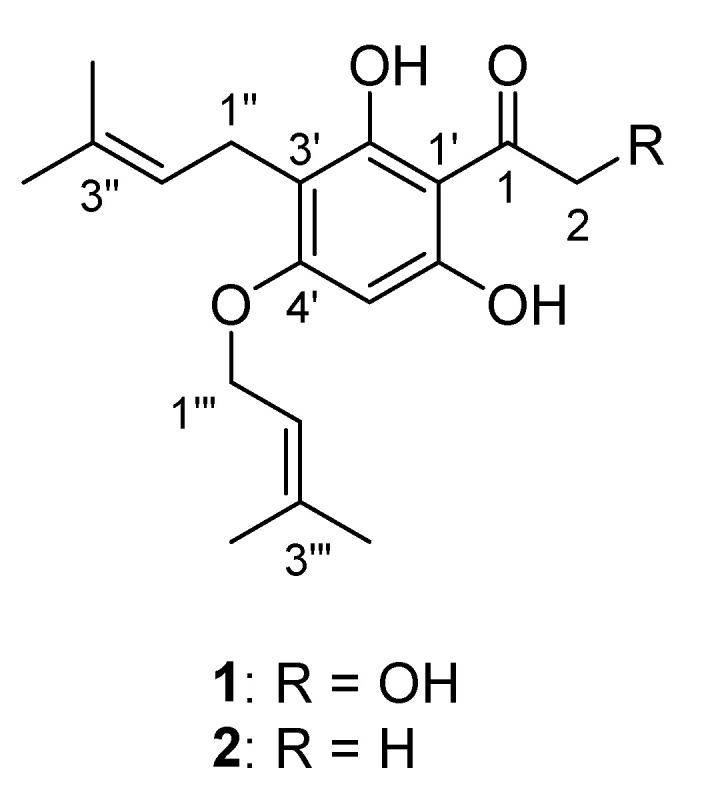
Structures of bioactive compounds isolated from the fruit of *A. crassipetala*.

**Figure 2 antibiotics-09-00487-f002:**
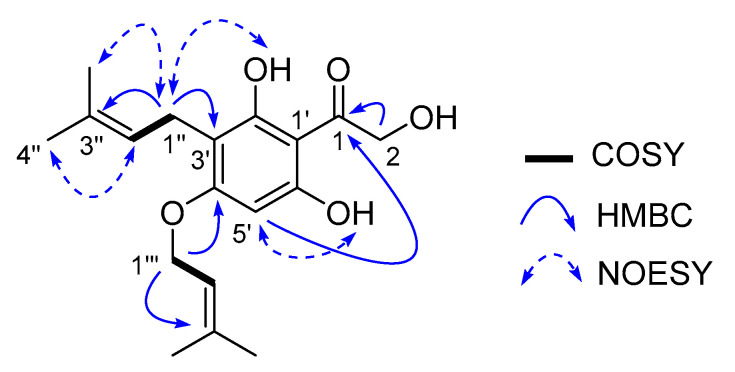
Key COSY, HMBC and NOESY correlations of crassipetalonol A (**1**).

**Table 1 antibiotics-09-00487-t001:** NMR Spectroscopic Data (^1^H 400 MHz, ^13^C 100 MHz) in DMSO-*d*_6_ for **1.**

Position	*δ* _C_	mult.	*δ*_H_ (*J* in Hz)	NOESY	HMBC
1	204.4	C			
2	68.2	CH_2_	4.63, s		1
1′	102.4	C			
2′	161.5	C			
3′	107.0	C			
4′	162.5	C			
5′	91.3	CH_2_	6.06, s	6′-OH, 1‴	1′, 3′, 4′, 6′, 1 *^b^*
6′	160.7	C			
1″	20.9	CH_2_	3.11, d (*J* = 7.2)	2′-OH, 5″	2′, 3′, 4′, 2″, 3″
2″	122.9	CH	5.07, t (*J* = 7.2)	4″	1″, 4″, 5″
3″	129.9	C			
4″	25.4	CH_3_	1.59, s	2″	2″, 3″, 5″
5″	17.5	CH_3_	1.67, s	1″	2″, 3″, 4″
1‴	64.6	CH_2_	4.52, d (*J* = 6.4)	5′, 4‴	4′, 2‴, 3‴
2‴	119.2	CH	5.40, t (*J* = 6.4)	5‴	4‴, 5‴
3‴	137.8	C			
4‴	18.0	CH_3_	1.70, s	1‴	2‴, 3‴, 5‴
5‴	25.5	CH_3_	1.75, s	2‴	2‴, 3‴, 4‴
2-OH			*^a^*		
2’-OH			13.45, s	1″	1′, 2′, 3′
6’-OH			10.97, s	5′	1′, 5′, 6′

*^a^* Not observed. *^b^* Weak signal.

**Table 2 antibiotics-09-00487-t002:** Antibacterial activity towards G+ve bacteria crassipetalonol A (**1**) and crassipetalone A (**2**).

Compound	MIC_75_ (µM) *^c^*				MBC (µM) *^d^*			
	*S. aureus* 29247	*S. aureus* 25923	*E. faecium* 35667	*E. faecium* c15	*S. aureus* 29247	*S. aureus* 25923	*E. faecium* 35667	*E. faecium* c15
**1**	*^a^*	78.1	*^a^*	*^a^*	*^a^*	*^a^*	*^a^*	*^a^*
**2**	5.1	2.6	20.6	20.6	20.6	20.6	20.6	*^a^*
Chloramphenicol	9.7	9.7	9.7	9.7	*^b^*	*^b^*	*^b^*	*^b^*

*^a^* Not active at the maximum tested concentration of 50 µg/mL (approximately 160 µM). *^b^* Not active at the maximum tested concentration of 100 µg/mL (approximately 310 µM). *^c^* Minimum inhibitory concentration required to inhibit the growth of 75% of bacteria. *^d^* Minimum bactericidal concentration.

**Table 3 antibiotics-09-00487-t003:** Cytotoxic evaluation for **1**–**2**.

Compound	IC_50_ (µM) *^a^*				
	HaCaT	HDF	NFF	HEK293	HepG2
**1**	15.8	16.7	29.1	13.4	21.3
**2**	8.5	6.4	13.3	8.6	9.7
Doxorubicin	0.010	0.060	0.360	0.006	0.430

*^a^* Half the maximal inhibitory concentration.

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
