# Peer review of "Potent Antibacterial Prenylated Acetophenones from the Australian Endemic Plant Acronychia crassipetala"

_antibiotics, 2020, doi:10.3390/antibiotics9080487_

Round 1

Reviewer 1 Report

Submitted manuscript has focused on characterisation of two compounds from Australian Endemic Plant Acronychia crassipetala. Overall, the ms is well written but I have some concerns about the methods and results:

  1. Determination of antibacterial activity of selected compounds. There is no information about the condition. Were the microplate shaken? or the plate was incubated under static conditions. The final volume in each well was 200 ul and at this volume the shaking will be not effective.
  2. I suggest to discuss the obtained results and compare the cytotoxicity with antibacterial efficacy. It seems that inhibitory concentration of compound 2 caused higher cytotoxicty to human cell line. Therefore, these compounds have no commercial/scientific values.
  3. Discussion need to be improved and authors need discuss their results with other literautre.

Author Response

Reviewer’s comments and our responses:

Reviewer 1

Comment 1: Determination of antibacterial activity of selected compounds. There is no information about the condition. Were the microplate shaken? or the plate was incubated under static conditions. The final volume in each well was 200 ul and at this volume the shaking will be not effective.

Response: Broth microdilution assays are one of the most common ways to assess antimicrobial susceptibility in the research and clinical environment. The method used in this paper has previously been published (Tran et al, Journal of Natural Products, 2019, 82, 2809-2817) and followed standard methodology (e.g. M07-A10 Methods for Dilution Antimicrobial Susceptibility Tests for Bacteria That Grow Aerobically; Approved Standard—Tenth Edition CSLI guidelines).  In some cases, these methods used shaking and in other cases no shaking, depending on the publication (the reviewer is correct in that shaking 200 ul will not result in agitation and aeration seen in larger volumes).  However, we have now stipulated in the methods section that the cultures in our study were not shaken (line 193). We have also included references 32 and 33 supporting our methodology.

Comment 2: I suggest to discuss the obtained results and compare the cytotoxicity with antibacterial efficacy. It seems that inhibitory concentration of compound 2 caused higher cytotoxicity to human cell line. Therefore, these compounds have no commercial/scientific values.

Response: The discussion of the cytotoxicity with antibacterial efficacy of compound 2 has been added (lines 134-139).

“Although the selectivity indices between human cancer cells and bacterial cells of compound 2 ranged from 1 to 5, which is relatively low, its potent inhibition against the growth of S. aureus compared to other prenylated acetophenones reported previously [26,29-31] warrants further investigation, including in vivo trials to confirm the value of this compound.  In addition, compound 2 could be modified using medicinal chemistry approaches with an aim to further improve the activity/toxicity window.”

Comment 3: Discussion need to be improved and authors need discuss their results with other literature.

Response: The discussion of antibacterial activity has been added (lines 113-126).

“Although activity against fungi and G+ve bacteria has previously been reported for acetophenone and its derivatives [24-26], the presence of the phenolic hydroxy groups with acidity resulted in increased biological activity by uncoupling oxidative phosphorylation [27]. Moreover, the hydrophilic/lipophilic balance of the molecule was found to play an important role in the penetration of the antibacterial agent through a bacterial cell surface [24]. A certain degree of lipophilicity produced by the iso-prenyl and other substituents in the acetophenone molecule enhanced the antimicrobial activity [26,27]. The higher lipophilicity of compound 2 compared to 1 was predicted by their octanol-water partition coefficient (ClogP) values (4.63 of 2 versus 3.77 of 1) [28]. Therefore, compound 2 could penetrate more easily through the cell wall and exert its bactericidal activity. This study also revealed that the isolated acetophenones selectively inhibited the growth of the tested G+ve bacteria rather than the Gram-negative (G-ve) ones (Table S1, Supplementary materials). These results were in accordance with previous reports of the antibacterial activity of related prenylated acetophenones [26,29,30]. The selective activity of 2 might be related to cell wall disruption or to another specific target present only in G+ve bacteria.”

Reviewer 2 Report

Add some recent references into discussion part and try to discuss better the biological activity of the two isolated compounds  and see also the details revision in the attacehd pdf file.

Author Response

Reviewer’s comments and our responses:

Reviewer 2

Comment 1:  Add some recent references into discussion part and try to discuss better the biological activity of the two isolated compounds.

Response: More references and discussion have been added; see response to comments 2 and 3 of review 1.

Comment 2:  Please differentiate well the two isolated compounds (line 18).

Response: The trivial names of compounds 1 (crassipetalonol A) and 2 (crassipetalone A) were suggested using the combination of the plant species name A. crassipetala from which the compounds were isolated and the functional groups their chemical structures possess (alcohol for 1 and ketone for 2). We adopted this system using similar examples from the literature.

Comment 3:  I think you mean "new" (line 21).

Response: Compound 2 is a known compound with information of previous isolation and identification mentioned in lines 102-106; see reference 21.

Comment 4: Rewrite “the anticholinergic and antispasmodic drug, scopolamine butylbromide” as it is not correct.

Response: The sentence has been re-written (lines 36-38).

“…scopolamine, which is an important precursor for the synthesis of scopolamine butylbromide, an anticholinergic and antispasmodic drug with a brandname Buscopan [5]”

Comment 5:  This key word was repeated two times.

Response: The names of the two compounds are correct in the original manuscript.

Comment 6:  Write the Author (twice).

Response: The authors have been added.

Duboisia myoporoides R. Brown; line 35-36.

Acronychia crassipetala T.G. Hartley; line 16, 49 & 155.

Elaeocarpus grandis F. Muell; line 39.

Fontainea picrosperma C. T. White; line 42.

Euodia lunu-ankenda T. G. Hartley; line 105.

Urtica dioica L.; line 105.

Comment 7:  It is not correct using "now" and "has been" in the same phrase, please correct the tense. (lines 42-43). Delete “is” (line 44)

Response: The sentence has been corrected (lines 43-45).

Tigilanol tiglate has been approved by the European Medicines Authority [8] as a novel canine therapy for mast cell tumours [9] and is currently in clinical trials to assess its potential as a human anticancer therapeutic [10]”.

Comment 8:  Use the abbreviations G+ve and G-ve.

Response: The abbreviations G+ve and G-ve have been added.

Comment 9:  Write the full scientific name in the first mention S. aureus, excluding the abstract.

Response: The full name of “S. aureus” has been given (line 50).

Comment 10:  Replace with "extracted" (line 61).

Response: “Distilled” has been replaced with “extracted” (line 62).

Comment 11:  Replace with "research" (line 63).

Response: “Paper” has been replaced with “research” (line 64).

Comment 12:  Selectively inhibited the growth of the tested G+ve bacteria rather than the G -ve ones (lines 113-114).

Response: The sentence has been corrected (lines 123-124).

Comment 13:  Please use the same tense, you used sometimes "suggested" and sometimes "suggest". it is not correct.

Response: The tense of the sentence has been corrected (lines 131-132).

“However, comparing the antibacterial and cytotoxic activities suggested that compound 2 had more potential as an antibiotic than compound 1”.

Comment 14:  Write the name of the two compounds in the caption.

Response: The name of the two compounds have been added in the caption of Table 2.

Comment 15:  IC50, please write "50" in the lower case.

Response: IC50 has been reformatted to IC50 (“50” is subscript) (line 143-44).

Comment 16:  Did you use the "Hexane" alone as control, because the antibacterial activity could be related to the "used organic solvent" not to the "extracted compounds".

Response: “The hexane extract” only refers to the extract obtained.  However, the hexane is removed by evaporation and all assays are subsequently completed using DMSO as the vehicle.

To clarify the preparation of the extracts, the following sentences have been added to section “3.3 Extraction and Isolation” (lines 163-165).

“The solvents were then evaporated to yield three extracts (hexane, DCM and MeOH). To 10 mg of each extract, 1 mL of DMSO was added to prepare a stock concentration of 10 mg/mL for MIC and MBC assays.”

Comment 17:  Is this period sufficient for the complete bacterial growth and production of bioactive metabolites ? How can confirm that? Did you try to draw the bacterial growth curve firstly.

Response: The protocol we used in the paper is standard in the microbiology field (e.g. M07-A10 Methods for Dilution Antimicrobial Susceptibility Tests for Bacteria That Grow Aerobically; Approved Standard—Tenth Edition CSLI guidelines). We have also published research using this method recently (Tran et al, Journal of Natural Products, 2019, 82, 2809-2817). The length of incubation (24 hrs) is therefore, well understood and is sufficient to support microbial growth and release of metabolites (positive growth was observed in our control well without antibiotic, or plant extract and inhibition was observed using a positive control antibiotic).  Longer incubation may in fact lead to nutrient deprivation and reduction in bacteria viability. We have provided references 32 and 33 to support our methodology.

Round 2

Reviewer 1 Report

Authors answered all raised issues. Revised version of manuscript was improved.

Reviewer 2 Report

The paper has been revised perfectly.